# Superconductivity from energy fluctuations in dilute quantum critical polar metals

**Pavel A. Volkov** [1] ✉, **Premala Chandra**[1] **& Piers Coleman** [1,2]

Superconductivity in low carrier density metals challenges the conventional electron-phonon theory due to the absence of retardation required to overcome Coulomb repulsion. Here we demonstrate that pairing mediated by energy fluctuations, ubiquitously present close to continuous phase transitions, occurs in dilute quantum critical polar metals and results in a dome-like dependence of the superconducting $T_c$ on carrier density, characteristic of non-BCS superconductors. In quantum critical polar metals, the Coulomb repulsion is heavily screened, while the critical transverse optical phonons decouple from the electron charge. In the resulting vacuum, long-range attractive interactions emerge from the energy fluctuations of the critical phonons, resembling the gravitational interactions of a chargeless dark matter universe. Our estimates show that this mechanism may explain the critical temperatures observed in doped $SrTiO_3$. We provide predictions for the enhancement of superconductivity near polar quantum criticality in two- and three-dimensional materials that can be used to test our theory.

Superconductivity exemplifies the dramatic effects of interactions in many-body quantum systems[1]. In conventional superconductors, electrons exploit the electron-phonon interaction to overcome the Coulomb repulsion[2,3] by producing a highly retarded attraction that pairs electrons. This process requires a large ratio between the Fermi and Debye energies $E_F/\omega_D >> 1$[4]. A challenge to this mechanism is posed by superconductivity in low carrier density metals near polar quantum critical points (QCPs). Such materials, typified by doped $SrTiO_3$ (STO)[5], exhibit bulk superconductivity down to carrier densities of order $10^{19}$ cm$^{-3}$, where the relevant phonon frequency exceeds the Fermi energy[5] by orders of magnitude. Yet despite this inversion of energy scales, experiments[6,7] indicate a conventional s-wave condensate, with a ratio of gap to transition temperature $2\Delta/T_c \approx 3.5$ in agreement with BCS theory[7].

Several theories have been advanced to explain superconductivity in polar metals using the conventional electron-phonon interaction[8] and its extension to include plasmons[9–12]. Recently, it has been proposed that the underlying polar quantum criticality is a key driver in the pairing[13–16]. However, this appealing idea encounters a difficulty, for the critical modes of a polar QCP are transverse optical (TO) phonons for which the conventional electron-phonon coupling vanishes at

low momenta[17–19]. Alternative phonon coupling mechanisms requiring spin-orbit coupling or multiband effects[20–22] (such as Dirac points[23,24]) have also been examined and possible superconductivity[23,25,26] has been discussed.

We reassess superconductivity in quantum critical polar metals guided by two key observations: first, that the strong ionic screening associated with the enhanced dielectric constant severely weakens the electronic Coulomb interaction (Fig. 1a); second, that in the absence of strong spin-orbit coupling the transverse optical phonon modes, decoupled from the electron charge, can be likened to dark matter, for like baryons in the cosmos, the electrons do not directly interact with the the intense background of zero-point dipole fluctuations. Furthermore like dark matter, the presence of the TO modes is only revealed to the electrons via their stress-energy tensor. In particular, the electrons interact with the energy density of the TO phonons. We model this coupling by the Hamiltonian[27–29]:

$$H_{En} = g \int d^3x \rho_e(\mathbf{x})(\mathbf{P}(\mathbf{x}))^2 \tag{1}$$

[1]Center for Materials Theory, Department of Physics and Astronomy, Rutgers University, Piscataway, NJ 08854, USA. [2]Department of Physics, Royal Holloway, University of London, Egham, Surrey TW20 0EX, UK. ✉e-mail: pv184@physics.rutgers.edu

**a**   **b**

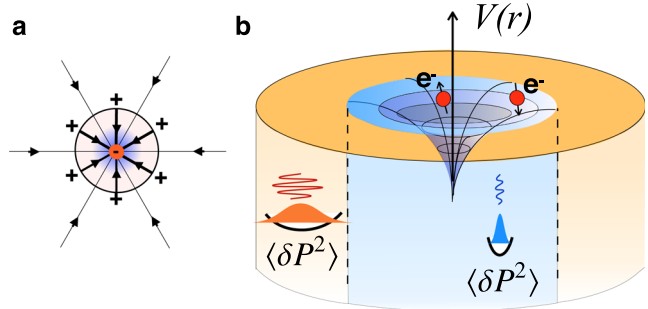

**Fig. 1 | Interactions between electrons in a quantum critical polar metal. a** the electric lines of force around an electron are ionically screened, **b** the fluctuations of the phonon energy density around electrons (see Eq. (1)) create an attractive potential well.

where $\rho_e(\mathbf{x}) = \psi^\dagger(\mathbf{x})\psi(x)$ is the electron density, $(\mathbf{P(x)})^2$ is proportional to the energy density of the local polarization $\mathbf{P}$ and $g$ is a coupling constant with the dimensions of volume. Microscopically, this interaction can be thought of as arising from the short-range effects of the Coulomb force within a unit cell of the material. The presence of an additional charge at the conduction electron site modifies the potential profile for the ions. A local increase in the electron density attracts the surrounding positively charged ions, reducing the distance between them. This causes the local "effective spring constant" of the phonons to rise in regions of high electon density. The natural units for this interaction are therefore atomic ones, i.e. unit cell volume, and should not be extremely different between different materials.

This coupling suppresses the zero-point fluctuations of the polarization in the vicinity of electrons, which in turn lowers the chemical potential of nearby electrons (Fig. 1b), creating an attractive potential well. To lowest order, the resulting attractive potential is described by the virtual exchange of pairs of TO phonons (see Supplementary Information for details), allowing us to link these ideas to two recent observations: first, that two-phonon exchange appears to drive the anomalous "high-temperature" $T^2$ resistivity of polar metals[29,30], and second that two-phonon processes may drive superconductivity[31], reviving an old idea[27]. At the same time, energy fluctuations are present in the vicinity of all continuous phase transitions, implying that their coupling to electrons must be relevant for a broad range of systems.

Here we study the consequences of the coupling to energy fluctuations of the order parameter (1) in quantum critical polar metals. We find that polar quantum criticality results in long-range "gravitational" interactions that mediate attraction between the electrons in an electromagnetically neutral background. This attraction overcomes the critically screened Coulomb repulsion and leads to superconductivity at extremely low densities. With increasing carrier concentration, the long-range character of the attraction is concealed due to shorter distances between electrons, leading to a dome-like superconducting region in the phase diagram. The application of our theory to SrTiO$_3$ shows good agreement with the observed magnitude and doping dependence of $T_c$. We predict that the impact of this novel mechanism will be considerably enhanced in two-dimensional quantum critical polar systems.

## Results

Our theory is built on an isotropic model for the polar metal, with an action $S[\psi^\dagger, \psi, P] = S_e + S_C + S_{En}$. Here $S_{En} = \int d\tau H_{En}$ is the energy fluctuation term (1), $S_e = \sum_k \psi_k^\dagger(\epsilon_\mathbf{k} - i\epsilon_n)\psi_k$, is the electronic action, in terms of the Fourier transformed electron field $\psi_k$, where $k \equiv (i\epsilon_n, \mathbf{k})$ is a four-vector containing the Matsubara frequency $\epsilon_n = (2n+1)\pi T$ and wave-vector $\mathbf{k}$. The action of electrostatic interactions and the polar

phonons is given by

$$S_C = \sum_q \left[ \frac{|e\rho_q - (\nabla \cdot \mathbf{P})_q|^2}{2\epsilon_0 \epsilon_1 \mathbf{q}^2} + \frac{\omega_n^2 + \omega_{T0}^2 + c_s^2 \mathbf{q}^2}{2\epsilon_0 \Omega_0^2} |\mathbf{P}(q)|^2 \right] \quad (2)$$

where the total charge densities $e\rho_e - \nabla \cdot \mathbf{P}$ and $q \equiv (i\omega_n, \mathbf{q})$, where $\omega_n = 2n\pi T$. $\epsilon_1$ is the bare dielectric constant, $\Omega_0$ is the ionic plasma frequency, $\omega_{T0}$ is the transverse optical mode frequency and $c_s$ is the speed of the transverse optical mode. $\omega_{T0}^2$ vanishes at the QCP. The Gaussian coefficients of the polarization, $\delta^2 S_C / \delta P_a(-q)\delta P_b(q) = D_{ab}^{-1}(q)$ in (2), separate into transverse and longitudinal components

$$D_{\alpha\beta}^{-1}(q) = D_L^{-1}(q)\hat{q}_\alpha \hat{q}_\beta + D_T^{-1}(q)(\delta_{\alpha\beta} - \hat{q}_\alpha \hat{q}_\beta) \quad (3)$$

where $D_{L,T}^{-1}(q) = (\omega_n^2 + \omega_{T0,L0}^2 + c_s^2 q^2)/\epsilon_0 \Omega_0^2$ are the inverse longitudinal and transverse phonon propagators. The longitudinal optical mode frequency $\omega_{L0}^2 = \omega_{T0}^2 + \Omega_0^2/\epsilon_1$ is shifted upwards by the Coulomb interaction.

We first consider the case where $g = 0$. Integrating over the longitudinal modes, we find that the Coulomb interaction becomes

$$\tilde{S}_C = \sum_q \left[ |\rho_q|^2 \frac{e^2}{2\epsilon_0 \epsilon(q)\mathbf{q}^2} + \frac{|\mathbf{P}^T(q)|^2}{2D_T(q)} \right], \quad (4)$$

where

$$\epsilon(\mathbf{q}, i\omega_n) = \epsilon_1 + \frac{\Omega_0^2}{\omega_n^2 + c_s^2 \mathbf{q}^2 + \omega_{T0}^2} \quad (5)$$

is the renormalized dielectric constant and $P_a^T(q) = (\delta_{ab} - \hat{q}_a \hat{q}_b)P_b(q)$ are the transverse components of the polarization. Most importantly, in action (4) the quantum critical transverse polar modes are entirely decoupled from the electronic degrees of freedom, motivating the dark matter analogy.

Normally, low carrier density metals are considered strongly interacting, for the ratio of Coulomb to kinetic energy, determined by $r_s = 1/(k_F a_B)$, where $k_F \sim n_e^{1/3}$ is the Fermi momentum and $a_B = \frac{4\pi\epsilon\hbar^2}{m^*e^2}$ the Bohr radius, is very large at low densities. However, in a quantum critical polar metal, the large upward renormalization of the dielectric constant, Eq. (5), severely suppresses the interaction between the electrons. Indeed, the dielectric constant at the relevant electronic scales at low densities is $\epsilon \sim \epsilon(\mathbf{q}, \omega_n)|_{q=2k_F, \omega_n=E_F} \approx \frac{\Omega_0^2}{(2c_s k_F)^2} \gg 1$ at the polar QCP, leading to $r_s \ll 1$. Furthermore, the electronic corrections to the dielectric constant, given in random phase approximation (RPA) by $\delta\epsilon_{RPA} = \frac{e^2}{q^2 \epsilon_0}\Pi_e(\mathbf{q}, \omega_n)$, where $\Pi_e(q, \omega_n)$ is the dynamical susceptibility (Lindhardt function) of the electron gas, can be neglected. Indeed, $\frac{\delta\epsilon_{RPA}}{\epsilon}|_{q=2k_F, \omega_n=E_F} \sim r_s \ll 1$.

The regime considered here lies in stark contrast to the conventional case, where the relevant longitudinal phonon frequency is much smaller than $E_F$. In this regime, the electronic plasmon appears at low energies $v_F q \ll \omega_n$. However, its contribution to pairing is suppressed by the factor $\epsilon^{-1}$[10]. Thus, in what follows we neglect this possibility, approximating the electron dynamical susceptibility by its long wavelength, low-frequency limit as in (5).

We next consider the effect of turning on the coupling to energy fluctuations in (1). The presence of a finite electron density $n_e = \langle \rho_e(x) \rangle$ leads to a shift in the phonon frequency:

$$\omega_T^2(n_e) = \omega_{T0}^2 + 2gn_e\epsilon_0\Omega_0^2, \quad (6)$$

which naturally explains the suppression of the polar state by charge doping, universally observed in polar metals[15,32]. Though this effect only shifts the position of the polar QCP, it must be included when

considering properties as a function of electronic density. For example a system that is initially polar with $\omega_{T0}^2 < 0$ and $g > 0$ will have a polar QCP at a finite density, as is observed in Ca-Sr substituted SrTiO$_3$[15].

The coupling of the energy fluctuations to the electron density fluctuations $\delta\rho_e(x) = \rho_e(x) - n_e$, cannot be integrated out exactly. Interactions with critical fluctuations near QCPs can be relevant perturbations in a scaling sense[33], destabilizing the Fermi liquid ground state already at weak coupling[33]. In our case, however, the interaction Eq. (1) preserves the Fermi liquid. Assuming the dynamical critical exponent $z = 1$ and taking the scaling dimension of momentum $[q] = 1$, one obtains $[g] = 2 - d$, irrelevant in 3D (see Supplementary Information for details), implying that the system remains a Fermi liquid even at the QCP (whereas in the known cases of non-Fermi liquid behavior the interaction remains at least marginal[33]). Thus, we can consider its effects perturbatively for weak coupling. Integrating out the field $\mathbf{P}(x)$ to lowest order in $g$, we obtain an effective interaction between electrons:

$$\Delta S = \frac{1}{2}\int d^4x\, d^4x'\, \delta\rho_e(x)V_{En}(x-x')\delta\rho_e(x') + O(g^3) \tag{7}$$

where

$$V_{En}(x-x') = -2g^2\mathrm{Tr}\left[D(x-x')^2\right] \tag{8}$$

is recognized to be an attractive density-density interaction resulting from two-phonon exchange, Fig. 1b. At criticality, the contribution to Eq. (8) of the transverse modes stems from their propagator

$$D_{ab}^{tr}(\mathbf{x},\tau) = \frac{\varepsilon_0}{c_s}\left(\frac{\Omega_0}{2\pi}\right)^2\frac{1}{\mathbf{x}^2 + c_s^2\tau^2}(\delta_{ab} - \hat{x}_a\hat{x}_b), \tag{9}$$

leading to an attractive long-range interaction of the form

$$V_{En}^{cr}(\mathbf{x},\tau) = -\frac{\alpha}{\left(\mathbf{x}^2 + c_s^2\tau^2\right)^2}, \tag{10}$$

where $\alpha = \left(\frac{g\varepsilon_0\Omega_0^2}{2\pi^2 c_s}\right)^2$, which plays the role of an emergent "gravitational" attraction between electrons at a polar QCP. These interactions also act on the TO phonons via the quartic term in their action[33,34] (see Supplementary Information for details). In what follows we will not consider this effects, assuming the relevant momenta and energies to be always larger than the critical region in the energy-momentum space set by the phonon-phonon interactions.

Away from criticality, (10) is valid for space-time separations smaller than the correlation length $\xi = c_s/\omega_T$. The qualitative form of the interaction at finite momentum and frequency transfer, relevant for pairing, is obtained by a Fourier transform of this expression with a long-distance cutoff $\xi$

$$\begin{aligned}V_{En}(i\omega_n,\mathbf{q}) &\approx -\frac{\alpha}{c_s^2}\int_{a_0}^{\xi}\frac{e^{i[(\mathbf{q}\cdot\mathbf{x})+\omega_n\tau]}}{x^4}d^4x \\ &\sim -\left(\frac{2\pi^2\alpha}{c_s}\right)\ln\left[\frac{a_0^{-1}}{\max(\xi^{-1},\omega_\mathbf{q}/c_s)}\right]\end{aligned} \tag{11}$$

where $\omega_q = \sqrt{\omega_n^2 + c_s^2\mathbf{q}^2}$. The characteristic electronic momentum and energy transfer scales are $q \sim k_F \sim n^{1/3}$ and $\omega_n \sim E_F \sim n^{2/3}$, respectively, resulting in $|q|^{-1} \sim n^{-1/3}$, $c_s/|\omega_n| \sim n^{-2/3}$. Furthermore, following (6), a finite electron density leads to a finite correlation length of the order $\xi \sim n^{-1/2}$. Consequently the interaction character changes with density, and can be described by an effective interaction $V_{En}^{Pair}(\mathbf{x},\tau)$ obtained by inverse Fourier transforming the final result of Eq. (11) (Fig. 2). For a low density polar metal that is critical at zero doping ($\omega_{T0} = 0$) one expects the

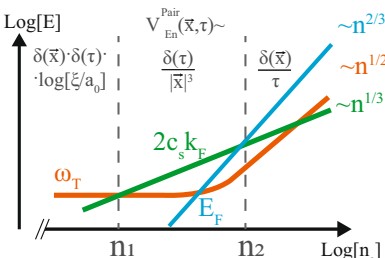

**Fig. 2 | Density-dependence of the form of the effective energy-fluctuation mediated electron-electron attraction $V_{En}^{Pair}(\mathbf{x}\tau)$, Eq. (11).** The colored lines show the three energy scales relevant for pairing: $\omega_T(n_e)$ (Eq. (6)), related to the polar order correlation length $\xi = c_s/\omega_T(n_e)$, the characteristic momentum-exchange scale $2c_s k_F$ and the Fermi energy $E_F$, the characteristic energy exchange scale. In each region (separated by gray dashed lines) the dominant scale determines the effective form of the interaction. At low densities $n_e \ll n_1 \approx \frac{1}{3\pi^2}\left(\frac{\omega_{T0}}{2c_s}\right)^3$ the interaction can be approximated with a local one. On increasing the density ($n_1 \ll n_e \ll n_2 = 3\pi^2\left(\frac{4m^*c_s}{\hbar}\right)^3$), the momentum dependence of the interaction becomes prominent first, such that the interaction can be approximated with an instantaneous, but non-local one (12). At the highest densities, strong retardation appears on the scale of order $\hbar/E_F$.

interaction to be cut off at the Fermi momentum $k_F$, while being essentially independent of frequency, since $E_F/c_s \ll k_F$. This is identical to an instantaneous repulsion, nonlocal in real space (Fig. 2, middle):

$$V_{En}^{Pair}(\mathbf{x},\tau)\big|_{c_s k_F \gg E_F,\omega_T(n_e)} \sim \frac{\delta(\tau)}{|\mathbf{x}|^3} \tag{12}$$

in analogy with the instantaneous Newtonian gravity emerging at low energies from the "relativistic" interaction (10). This case is also realized for a system away from a QCP ($\omega_{T0} > 0$) at intermediate densities ($c_s k_F \gg \omega_T(n_e), E_F$, see Fig. 2). In the high density limit, the frequency-dependence of the interaction can be no longer ignored and it is suppressed for frequencies beyond $c_s k_F$, qualitatively similar to a usual phonon-mediated attraction (Fig. 2, rightmost region). Finally, if $\omega_T(n_e) \gg E_F, c_s k_F$ (leftmost region of Fig. 2), which can be realized away from the QCP at the lowest densities, the interaction $V_{En}^{Pair}(\mathbf{x},\tau)$ reduces to an instantaneous local attraction.

A more detailed calculation of the interaction potential (see Supplementary Information for details) yields

$$\begin{aligned}V_{En}(i\omega_n,\mathbf{q}) &= -\left(\frac{2\pi^2\alpha}{c_s}\right)\left(\log\left[\frac{\Omega_T}{\omega_T(n_e)}\right] - f\left(\frac{\omega_q}{\omega_T(n_e)}\right)\right), \\ f(x) &= \frac{\sqrt{4+x^2}}{2x}\log\left[\frac{\sqrt{4+x^2}+x}{\sqrt{4+x^2}-x}\right] - \frac{1}{2},\end{aligned} \tag{13}$$

where $\Omega_T$ is the upper cutoff for the optical phonon frequency throughout the Brillouin zone. Here it is important to note, that since the integral is logarithmically divergent, large momenta of the order of the Brillouin zone size contribute significantly. At such momenta, the dispersion is expected to deviate from simply quadratic. Thus in practical calculations, an average value of $c_s$ should be used in the prefactor of (13), which we take in the spirit of Debye approximation to be equal to $\bar{c}_s = \Omega_T/((6\pi^2)^{1/3}/a_0)$. We remark that the contribution of the longitudinal modes to the induced interaction can be shown to be negligible in the critical low-density regime, where $\Omega_T \gg \omega_T(n_e), c_s k_F$ and $\frac{\Omega_0^2}{c_s^2 k_F a_B^{-1}} \gg 1$ (see Supplementary Information for details).

## Superconductivity at low densities

We now show that the attractive interaction mediated by the energy fluctuations always leads to superconductivity at low densities close to the polar QCP. In particular, the density is assumed to be in the middle

regime of Fig. 2, i.e. $E_F, \omega_T(n_e) \ll 2c_s k_F$, allowing us to neglect $\omega_n$ in the interaction. Averaging the repulsive Coulomb ((4), (5)) and the attractive (13), interactions over the Fermi surface, i.e. $\langle V(k - k') \rangle = \langle V(k_F, k_F, \theta) \rangle_\theta$, we obtain (see Supplementary Information for details):

$$\lambda = N(0) \left( \left( \frac{2\pi^2 \bar{\alpha}}{\bar{c}_s} \right) \left\{ \log \left[ \frac{\Omega_T}{2c_s k_F} \right] + 1 \right\} - \frac{(e^2/\varepsilon_0) c_s^2}{\Omega_0^2} \right), \quad (14)$$

where $N(0) = \frac{k_F m^*}{2\pi^2 \hbar^2}$ is the density of states and $\bar{\alpha}$ is equal to $\alpha$ with $c_s \to \bar{c}_s$. In deriving this we used that $\omega_T \sim (gn)^{1/2}$, $E_F \sim n^{2/3} \ll c_s k_F \sim n^{1/3}$. Most importantly, we find that at low enough doping the two-phonon attraction inevitably overcomes the Coulomb repulsion due to the logarithmic enhancement of the former. In particular, the total interaction is attractive for densities below

$$n_{cr} = \frac{1}{4 a_0^3} e^{-3\gamma}, \quad \gamma = \frac{128 \pi^5 c_s^2 \bar{c}_s^3 e^2}{\Omega_0^6 g^2} \quad (15)$$

where $\gamma \sim 40 \frac{a_B}{a_0} \frac{E_h \Omega_T^5}{\Omega_0^6}$ for $g \sim a_0^3$. For strongly polar materials, $\Omega_0 \gg \Omega_T$ and hence $\gamma \ll 1$, so this restriction is unimportant. Moreover, from (14) the attractive part of the interaction behaves as $\lambda \propto k_F \{ \log(\Omega_T/(2c_s k_F)) + 1 \}$, describing a dome-shaped behavior of the attractive coupling constant peaked at $k_F = \Omega_T/2c_s$ corresponding to a density:

$$n_{max} = \frac{1}{3\pi^2} \left( \frac{\Omega_T}{2c_s} \right)^3 \approx \frac{(\bar{c}_s/c_s)^3}{4 a_0^3}. \quad (16)$$

As the phonon dispersion flattens near the Brillouin zone edges, the average $\bar{c}_s < c_s$ so that $n_{max} a_0^3 \ll 1$, corresponding to a dilute charge concentration below half filling. Finally, away from the QCP (i.e. $\omega_{T0} \neq 0$), the Coulomb screening is reduced, resulting in an additional repulsive term $\sim 2\pi e^2 \omega_T^2(n_e)/(\Omega_0^2 k_F^2)$ (c.f. (5)). Due to its singular nature at $k_F \to 0$ this sets a lower bound on the density $n_{min} \sim \xi^{-3} [\gamma/\log(\Omega_T/\omega_T(n_e))]^{3/2}/3\pi^2$ where the interaction is attractive. Therefore, both proximity to the QCP and sufficiently low densities are required to generate an attractive interaction.

Let us now discuss the critical temperatures of the resulting superconductor. At low densities, the interaction is essentially instantaneous (see Fig. 2). The critical temperature can then be found in the non-adiabatic weak-coupling limit to be $T_c \approx 0.28 E_F e^{-1/\lambda}$ (which includes vertex corrections to the interaction)[35,36]. Due to the exponential dependence on the coupling constant, one expects $T_c$ to have a dome-like shape with a maximum at $n_{max}$ as in (16). The theory developed here also has important consequences for the dependence of $T_c$ on external tuning parameters (e.g. pressure) in the vicinity of a polar QCP. Neglecting the residual Coulomb term in (14) and thus assuming the dominance of the energy-fluctuation attraction, one obtains

$$\frac{d \log T_c}{d \log \omega_T(n_e)} \propto \begin{cases} -\frac{1}{\log^2 \frac{\Omega_T}{\omega_T(n_e)}}, & \omega_T(n_e) \gg c_s k_F \\ -\frac{\omega_T^2(n_e) \log\left(\frac{c_s k_F}{\omega_T(n_e)}\right)}{(c_s k_F)^2}, & \omega_T(n_e) \ll c_s k_F. \end{cases} \quad (17)$$

For $\omega_T(n_e) \gg c_s k_F$, a singular dependence of $T_c$ on tuning parameter near the QCP is observed. However, at intermediate densities ($\omega_T(n_e) \ll c_s k_F$, Fig. 2, middle panel), the coupling constant is almost independent of the TO phonon frequency, so the tuning sensitivity will be much weaker.

## 2D polar metals
Similar calculations can be performed in two dimensions. While at tree level, Eq. (1) is marginal, the corrections due to quartic interactions

between phonons introduce an anomalous dimension to the energy fluctuations, reducing their momentum-space singularities[37], which allows to preserve the Fermi liquid in 2D (see Supplementary Information for details). We note that the anomalous dimension of the energy fluctuations makes them distinct from the perturbative two-phonon processes in the 2D case at low momenta/frequencies. Indeed, the presence of an anomalous dimension indicates that the phonons are no longer well-defined quasiparticles[33] so that the energy fluctuations can no longer be uniquely defined as a two-phonon exchange. However, for momenta larger than a critical one (set by the phonon-phonon interactions), (11) is expected to be valid. The 2D Fourier transformation of expression (11) yields

$$V_{En}^{2D}(i\omega_n, \mathbf{q}) \sim \frac{g_{2D}^2 \varepsilon_0^2 \Omega_0^4}{4\pi c_s^2} \frac{1}{\max(\omega_T(n_e), \omega_{\mathbf{q}})} \quad (18)$$

- a stronger singularity than in 3D. In the limit $c_s k_F \gg \omega_T(n_e)$ the terms due to LO phonon energy fluctuation also have to be included, being of the same order, but this does not change the qualitative form of the interaction (see Supplementary Information for details). Finally, the bare Coulomb repulsion in 2D is given by $\frac{e^2}{2\varepsilon_0 q}$. Screening with the polar mode and conduction electrons, however, reduces it to

$$V_C^{2D}(i\omega_n, \mathbf{q}) = \frac{\varepsilon_0^{-1}}{\frac{\Omega_0^2 l_0 \mathbf{q}^2}{\omega_n^2 + \omega_T^2(n_e) + c^2 \mathbf{q}^2} + 4 \frac{m^* e^2}{\hbar^2}}, \quad (19)$$

where $l_0$ is the 2D layer thickness.

## Superconductivity in Doped SrTiO$_3$
We now apply these results to doped SrTiO$_3$. Figure 3a displays the doping dependence of $T_c$ calculated using the parameters from the literature (see Supplementary Information for details) and taking the coupling $g$ as a fitting parameter. The resulting value $g/a_0^3 \approx 0.68$ is comparable to ones obtained from fits to the $T^2$ resistivity ($g/a_0^3 \approx 0.5$, note the factor of two in the definition of the coupling[29]) and doping dependence of the TO phonon frequency ($g/a_0^3 \approx 0.56$ can be deduced from[38]). We assume an effective one-band model with mass $m^* = 4m_e$. The Hall effect and quantum oscillation data[39] indeed suggest that one of the three bands occupied in SrTiO$_3$ contains the most carriers (by an order of magnitude) and has the largest effective mass, dominating the density of states. For the energy fluctuation interaction, Eq.(13), we assumed $2c_s k_F \gg E_F$, which holds in SrTiO$_3$ for densities lower than $2.6 \cdot 10^{19}$ cm$^{-3}$. As discussed before (see Fig. 2), this allows us to approximate the interactions with an instantaneous one. In this approximation, $2\Delta/T_c$ takes the BCS value 3.52[35,36], in accord with STM experiments[7].

If the Coulomb repulsion (represented for intermediate densities by the second term in Eq. (14)) is neglected (gray dashed line in Fig. 3a) both the magnitude and the doping dependence of the critical temperature are in good agreement with experiment[39,40] at all electron concentrations. The dome-like shape of $T_c(n_e)$ arises from the non-omonotonic dependence of the two-phonon attractive coupling constant on density, initially rising with the density of states, subsequently decreasing as the lower energy cutoff $c_s k_F$ grows. Importantly, the position of the $T_c$ peak, $n_{max}$ is approximately independent of the only free parameter of our theory, the coupling constant $g$ - indeed, Eq. (16) yields $n_{max} \approx 5.5 \cdot 10^{20}$ cm$^{-3}$, close to the actual value. Including the Coulomb repulsion (black line in Fig. 3a, see Supplementary Information for details), the results based on the instantaneous approximation do not agree as well with the experimental data for densities higher than $2.6 \cdot 10^{19}$ cm$^{-3}$. Indeed, since in this regime $E_F \gtrsim 2c_s k_F$, frequency dependence of the interaction can be no longer ignored. Whilst a detailed analysis of the frequency-dependent equations is beyond the scope of the current work, we observe that while the Coulomb

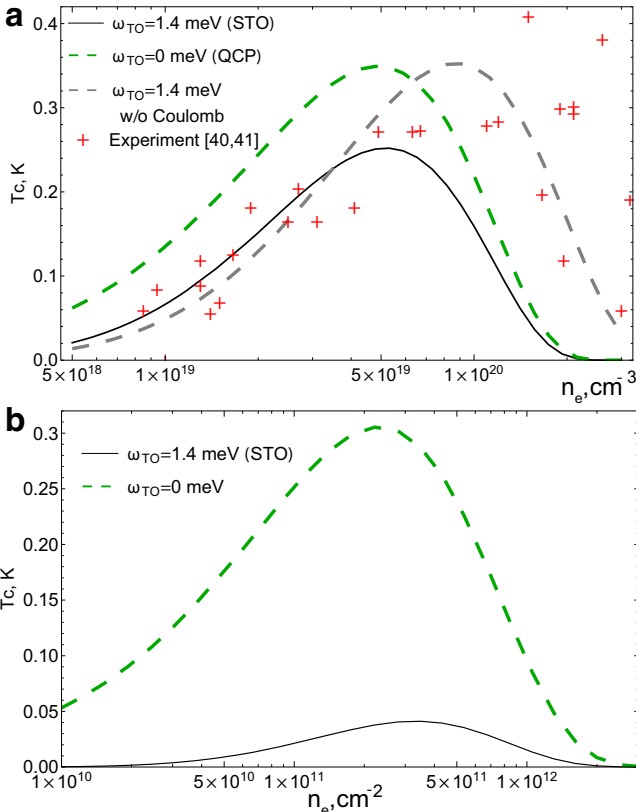

**Fig. 3 | Superconducting $T_c$ for the energy-fluctuation mechanism as a function of electron density. a** Black solid line: $T_c$ as a function of carrier density for parameters appropriate for SrTiO$_3$, where the best fit is obtained for $g/a_0^3 = 0.68$. The experimental $T_c$ is determined from the onset of the Meissner effect[39,40]. The green line shows the $T_c(n_e)$ expected at the QCP. The gray dashed line shows the best fit for $T_c$ ($g/a_0^3 = 0.62$) if the effects of Coulomb repulsion are ignored. **b** Same as (**a**) for a 2D system modeling a film of SrTiO$_3$ with thickness $2a_0$. $m^* = 1.8 m_e$ is taken due to low densities[41]. The enhancement due to proximity to the QCP (green line) is much stronger, than in the 3D case (**a**).

repulsion strengthens as a function of frequency (see Eq. (5)), the attraction from energy fluctuations, Eq. (11), becomes weaker. Thus we expect that the interaction will change its sign as a function of frequency, allowing for an enhancement of the pairing via the effects of retardation[2,3]. We note that a second superconducting dome has been observed at yet lower densities $n_e \sim 10^{18}$ cm$^{-3}$[41,42]; however, the Meissner effect is absent for this second dome[5,42]. This suggests a highly inhomogeneous state likely affected by the nature of the dopants[43], effects that lie beyond the current model.

As discussed above, proximity to a polar QCP should enhance $T_c$, particularly at low densities. Such a correlation has been observed for the cases of oxygen isotope substitution[14] as well as Ca-Sr substitution[15], pressure[11] or strain[44,45]. In particular, the enhancement is observed to be more pronounced at low dopings[15], in qualitative agreement with the arguments given above, see also Fig. 3a, green dashed line. Note also, that in the polar phase close to the QCP the interaction (1) would still lead to pairing due to fluctuations of the order parameter amplitude around the equilibrium value.

Above we have mentioned, that critical effects for pairing should be emphasized in 2D systems. We propose that thin films of SrTiO$_3$ may provide a platform to explore the predictions of the energy fluctuation theory. Figure 3b displays the predicted $T_c$, using $T_c = 0.15 E_F e^{-1/\lambda}$ for 2D instantaneous pairing ($c_s k_F \gg \omega_T(n_e)$)[36], using coupling constants appropriate for a two-layer thick slab of SrTiO$_3$. Assuming that the electrons and phonons are in the lowest lateral quantization state,

we obtain $g_{2D} = g/(2a_0)$ (see Supplementary Information for details). The results show that an appreciable $T_c$ is obtained at low densities. Furthermore, $T_c$ is highly sensitive to the approach to criticality: a slight decrease of the TO phonon frequency leads to an enormous increase of $T_c$.

## Discussion

New polar metals are being actively discovered[46]; our theory suggests that suppression of the polar structural transition, for example by pressure[11], promises to reveal a number of new low-density superconductors. In particular, perovskites BaTiO$_3$ and KTaO$_3$ can be tuned to ferroelectric QCPs by pressure[47] and strain[48], respectively, and become metallic under doping[32,49,50]. Additionally, two-dimensional systems created from these materials, such as LaAlO$_3$/SrTiO$_3$ interfaces[51] and surfaces of KTaO$_3$[52] crystals have been shown to exhibit superconductivity. In the former case, the doping dependence and maximal value of $T_c$ suggest that the pairing is driven by the same mechanism as in bulk SrTiO$_3$[53]. Also, a recently discovered intrinsic polar metal LiOsO$_3$[54] has been predicted to host nodal points close to the Fermi level (which suggests that small Fermi surfaces with dilute concentrations can be present or created by doping)[55] and to have a strain-tuned polar QCP[56]. In all of these cases, the role of energy fluctuation mechanism in pairing can be verified by the characteristic dependence of the phonon energy on electron concentration (Eq. (6)), scaling of $T_c$ with phonon energy (Eq. (17)), and the presence of a related $T^2$ component in resistivity at low densities[28,29]. At the same time, unlike typical scenarios of quantum critical pairing[57], the normal state is expected to be a Fermi liquid.

Intriguingly, the "dark matter" scenario identified here in polar metals indeed has a counterpart in the fermionic superfluid scenarios of cosmological dark matter[58,59], suggesting that quantum critical polar metals may provide a solid-state platform to study these proposals experimentally. In a broader context, energy fluctuations are ubiquitous at quantum critical points, and may thus play a role in electron pairing in many settings. One instance is the case of superconductivity occurring close to a nonpolar structural transition, driven by a phonon at a high-symmetry point, exemplified by tungsten bronze[60–62] and the A15 compounds[63]. As in a polar metal, the critical phonon decouples from electrons at low momenta, raising the intriguing possibility of energy-fluctuation pairing. Energy fluctuations have been also proposed as alternative drivers of pairing near antiferromagnetic quantum critical points in strongly correlated metals[64,65]. Finally, the phenomenon of superconductivity mediated by energy-density fluctuations can also serve as a tool to probe "dark matter" aspects of the solid state, involving excitations that do not interact electromagnetically, such as those related to complex hidden orders[66].

**Note Added:** While this work has been under review, Ref. 67 appeared, where a similar mechanism has been applied to SrTiO$_3$ at $n_e \lesssim 10^{18}$ cm$^{-3}$.

## Data availability

All data needed to evaluate the conclusions in the paper are present in the paper and/or the Supplementary Information. Additional data related to this paper may be requested from the authors.

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

## Acknowledgements

The authors gratefully acknowledge stimulating discussions with D. L. Maslov, and with the late P. W. Anderson during early stages of this work. P. A. V. is supported by a Rutgers Center for Materials Theory Abrahams Fellowship, P. Chandra is supported by DOE Basic Energy Sciences grant DE-SC0020353 and P. Coleman is supported by NSF grant DMR-1830707.

## Author contributions

P.A.V. contributed to formulating the problem, performed the analytical and numerical calculations and contributed to analyzing the results and writing the manuscript. P. Chandra contributed to formulating the problem, analyzing the results and writing the manuscript. P. Coleman contributed to formulating the problem, analyzing the results and writing the manuscript.

## Competing interests

The authors declare no competing interests.
