## [Peer Review File · Nature Communications]

Superconductivity from Energy Fluctuations in Dilute Quantum Critical Polar MetalsEditorial Note: This manuscript has been previously reviewed at another journal that is not operating a transparent peer review scheme. This document only contains reviewer comments and rebuttal letters for versions considered at *Nature Communications*.

Reviewer #1 (Remarks to the author):

The authors built a low-energy effective model in an attempt to study dilute quantum critical polar metals. The motivation is that in dilute limit, the Fermi surface is small, which constrains the phase space and leads to negligible direct coupling between itinerant electrons and polar phonon modes (which are transverse). Therefore the next available coupling is:

$$H_{En} = g \int dx^3 \rho_e(\mathbf{r}) (P(\mathbf{r}))^2 \quad (1)$$

where $\rho_e(\mathbf{r})$ is the charge density operator and $P(\mathbf{r})$ is the local polarization. The fluctuation of the charge density around its equilibrium value

$$\delta\rho_e(\mathbf{r}) = \rho_e(\mathbf{r}) - n_e$$

can lead to an effective attractive interaction in the perturbation expansion (the first term is proportional to g^2):

$$V_{En}(i\omega_n, \mathbf{q}) = - \left(\frac{2\pi^2\alpha}{c_s} \right) \left(\log \left(\frac{\Omega_T}{\omega_T} \right) - f \left(\frac{\omega_q}{\omega_T} \right) \right) \quad (2)$$

This attractive interaction has the origin of two-phonon exchange. At low carrier density, this attractive interaction will overcome Coulomb repulsion and lead to possible superconductivity. An effective electron-phonon coupling λ is:

$$\lambda \sim k_F \left[\log \left(\frac{\Omega_T}{2c_s k_F} \right) + 1 \right] \quad (3)$$

which has dome-like dependence on k_F . Since $k_F \sim n_e^{1/3}$ and the superconducting transition temperature $T_c \sim E_F \exp(-1/\lambda)$. Therefore, T_c has a dome shape as a function of carrier density n_e . The authors applied this theory to doped SrTiO₃.

Eq. (1) has been studied in literature (e.g. PRL 126, 076601 (2021)) and the resulting attractive interaction that arises from two-phonon exchange has been proposed as possible mechanism to explain superconductivity of doped SrTiO₃ (e.g. PRL 32, 215 (1974) and PR Research 1, 013003 (2019)). The novelty in this study is that the authors find that the charge fluctuation in Eq. (1) can lead to this attractive interaction and thus justify the two-phonon exchange as the pairing mechanism for superconductivity in dilute metals.

However, I have a few comments/questions for the authors to address:

1. The theory is supposed to describe “dilute quantum critical polar metals”. What parameter is tuned in order to achieve the polar quantum critical point (QCP)? In experiments of $\text{Ca}_x\text{Sr}_{1-x}\text{TiO}_{3-\delta}$ (NP 13, 643 (2017)) and doped BaTiO_3 (PRL 104, 147602 (2010)), doped electrons suppress the polar displacements and lead to a polar QCP at a finite carrier density n_e . However, in the current theory, the transverse polar phonon mode has dependence on n_e as:

$$\omega_T(n_e) = \sqrt{\omega_{T0}^2 + gn_e\epsilon_0\Omega_0^2} \quad (4)$$

where $\omega_{T0} > 0$ is the transverse polar phonon frequency when mobile carriers vanish. From Eq. (4), does that mean $g < 0$ so as to achieve a polar QCP at a finite n_e ? However, if $g < 0$, does that imply Eq. (1) may lead to some instability when $|P(\mathbf{r})|$ is sufficiently large? If $g > 0$, then where is the polar QCP? I suppose that $\omega_{T0} > 0$ is a constant throughout the calculations. Clarification is needed here.

2. In Fig. 2, the authors show that the effective pairing potential V_{En}^{Pair} may have three different forms as the carrier density n_e changes. For low carrier density (middle region),

$$V_{En}^{Pair}(\mathbf{r}, \tau) \sim \frac{\delta(\tau)}{|\mathbf{r}|^3} \quad (5)$$

For high carrier density (rightmost region),

$$V_{En}^{Pair}(\mathbf{r}, \tau) \sim \frac{\delta(\mathbf{r})}{\tau} \quad (6)$$

For intermediate density (leftmost region),

$$V_{En}^{Pair}(\mathbf{r}, \tau) \sim \delta(\mathbf{r})\delta(\tau) \log(\xi a_0) \quad (7)$$

There are a few confusing points that need to be clarified:

- * the x -axis of Fig. 2 is monotonic with the carrier density n_e . Then why does the leftmost region correspond to the “intermediate density”?
- * the paper wrote “For a low density polar metal that is critical at zero doping $\omega_T(n_e = 0) = 0$ ”, that means $\omega_{T0} = 0$. Is that a necessary condition to derive Eq. (5)? What happens if $\omega_{T0} > 0$?

- * For a finite $\omega_{T0} > 0$, since $2c_s k_F \sim n_e^{1/3}$ and $E_F \sim n_e^{2/3}$, when n_e is sufficiently small, $\omega_T \rightarrow \omega_{T0}$, which must exceed $2c_s k_F$ and E_F . Then does Eq. (7) correspond to ultralow densities, rather than “intermediate density”?

3. When the authors apply their theory to doped SrTiO₃, can they estimate what the two critical carrier concentrations are in Fig. 2, given the experimental information (e.g. c_s, ϵ_1)? If it is doped BaTiO₃, whether these two critical carrier concentrations will be very different?

4. While Eq. (1) is allowed by symmetry, what is the microscopic origin of this term? In particular, what does g depend on? Is it strongly material-dependent? For example, in doped SrTiO₃, the authors use $g/a_0^3 = 0.62$. Then how about doped BaTiO₃ or other doped ferroelectric semiconductors? A more general question is: what is the condition under which Eq. (1) becomes the dominant interaction in addition to Coulomb repulsion? Is it that the material must be close to the polar QCP? Or is it that the carrier density must be low? Or both?

5. The presentation needs a substantial improvement. This is a pure theoretical modelling. However, the notations are sometimes confusing. An incomplete list include:

- * In the main text, Eq. (11), (13), (18), what does $i\omega$ mean? Is ω a real frequency or a Matsubara frequency? Eq. (17) uses $i\omega_n$. Do they mean the same thing?
- * In the main text, Eq. (16) n_{max} has the dimension $1/[\text{length}]^3$. Then what does the text $n_{max} \ll 1$ mean?
- * In the main text, we have n_e and n . Do they mean the same thing (i.e. carrier density)? If yes, can we just use one symbol to avoid confusion?
- * In the main text, we have $\omega_T(\mathbf{q})$, $\omega_T(n_e)$ and a constant ω_T . In the SI, we have $\omega_T(n_e, \mathbf{q})$. In the derivation, we have $\omega_T \gg \omega_n$. What does that ω_T mean? It means the constant term (i.e. $n_e = 0$ and $\mathbf{q} = 0$), or a more general $\omega_T(n_e, \mathbf{q})$?
- * In the SI, Eq. (1) reads $S = S_e + S_{ph} + S_{e-ph}$. However, S_{e-ph} is not defined at all. Why not reads $S = S_e + S_{ph} + S_{En} + S_{Coul}$ explicitly?

* Sometimes a vector is denoted as an arrow on top of a letter and in other occasions is denoted as bold font. Is there a particular reason for this?

6. Finally I have a comment: the so-called “dilute quantum critical polar metal” is very rare. Probably doped SrTiO_3 is the only known example. SrTiO_3 is unique is that the material by itself is in the vicinity of ferroelectric phase transition. In other polar metals (such as LiOsO_3) or doped ferroelectric materials (such as doped BaTiO_3), while it is possible to drive them close to a polar QCP via strain or doping, usually the carrier density around the polar QCP is reasonably high ($\sim 10^{21}/\text{cm}^3$) and therefore the Fermi surface is sufficiently large so that the conventional electron-phonon mechanism is working. Therefore I am wondering whether the current pairing mechanism can be applied to any material system other than doped SrTiO_3 ?

Reviewer #2 (Remarks to the Author):

The authors of the present manuscript study the possibility of low-density superconductivity originating from the quadratic coupling between electron's density and soft transverse optical phonons. First, they study the effective interaction mediated by such two-phonon processes and show that it is attractive and has non-trivial dependence on electron's density. At sufficiently low density, such attraction overcomes the residual Coulomb repulsion which is significantly screened by the longitudinal optical mode. Then, the authors study the superconducting instability caused by the two-phonon processes in systems with parabolic dispersion. They show that the superconducting transition temperature as a function of electron's density has a dome-like shape. The initial enhancement of T_c with density is explained by the growth of the density of states, while the subsequent decrease is due to the suppression of the effective attraction with increasing density. Finally, the authors apply their results to explain superconductivity in strontium titanate.

The study is timely, while the topic and the results are very interesting. Although I think the paper deserves publication in some form, I believe the presentation can be improved significantly. I'll try to list the most important aspects that bothered me:

1. The first important step in your study is the calculation of the effective electron-electron attraction mediated by the two-phonon processes at different dopings. Depending on the doping and the proximity to the critical point, the behavior is very different. Could you summarize all the results for different regimes in a concise reader-friendly manner? I believe Fig. 2 is supposed to help with that but I find it hard to understand its meaning properly. For example, what are the densities that correspond to the vertical grey dashed lines on that figures? Is there a regime corresponding to Eq. (10)? When you compare ω_T to $c \cdot k_F$, do you have in mind the bare TO phonon mass or the one renormalized according to Eq. (6)? What's the meaning of different curves in Fig. 2?
2. You have introduced too many similar notations having different meaning. It makes it really hard to navigate across the paper. For example, there's ω_T , $\omega_T(n_e)$, $\omega_T(q)$, ω_q , Ω_T , Ω_0 all meaning different things.
3. For some reason, you spend a lot of time discussing the analogy with the "dark matter". Of course, we all love analogies between different concepts in physics, but this one seems a bit inappropriate. I'm not an expert in "dark matter", but I don't think people already know convincingly what that is and how exactly it interacts with more conventional matter. To me, the problem of the two-phonon mediated superconductivity is interesting enough on its own, just from the condensed matter perspective. Moreover, sentences like "We find that polar quantum criticality results in long-range "gravitational" interactions that mediate attraction between the electrons in an electromagnetically neutral background." almost made me believe that the nature of this interaction is gravitational rather than electromagnetic, which is extremely confusing.
4. Instead of discussing the relation to the "dark matter", I'd suggest to discuss the similarities and differences with some other works on the same topic. Specifically, it'd be very interesting to see some comments regarding this recent preprint: <https://arxiv.org/abs/2106.09530>. The authors of that work also obtained some log enhancement of the effective interaction in the Cooper channel, see their Eqs. (7)-(8), which, however, seems different from your log in Eq. (14), since it doesn't depend on k_F (on electron density). I understand that this paper appeared after yours, but nevertheless I'd be happy to see your comments.
5. You state above Eq. (7) that the Fermi liquid state is not destroyed even at the QCP based on the observation that the two-phonon coupling is irrelevant. But is it a sufficient

condition? Can it happen that some loop diagrams lead to some crucial (possibly non-analytic) corrections which eventually destroy the Fermi liquid at the QCP (like, e.g., Landau damping in a more conventional coupling to a critical mode)? I'm not encouraging you to perform such an analysis in this paper, but rather curious whether this statement is sufficiently substantiated.

In addition, I have some minor more technical questions and comments:

6. The second paragraph of the introduction mentions the alternative mechanism for coupling to the TO mode through the spin-orbit coupling. Then, the last sentence of this paragraph, "However, this appealing idea encounters a difficulty, for the critical modes of a polar QCP are transverse optic (TO) phonons that decouple from the electrons at low momenta", looks a bit strange and confusing since it is exactly the spin-orbit coupling that allows to couple electrons to the TO phonons directly even at zero momentum. Also, it would be appropriate to mention here some earlier papers that discussed that type of coupling and its consequences for superconductivity: <https://journals.aps.org/prb/abstract/10.1103/PhysRevB.74.012501>, <https://journals.aps.org/prl/abstract/10.1103/PhysRevLett.115.207002>, <https://journals.aps.org/prx/abstract/10.1103/PhysRevX.9.031046>

7. The first paragraph of the intro says "A challenge to this mechanism is posed by superconductivity in low carrier metals near polar quantum critical points (QCPs)." I believe it's meant to be "... in low carrier density metals..."

8. Above Eq. (2), the term S_e should contain the sum over k , not just \vec{k} (i.e., sum over frequencies is missing).

9. In Eq. (3), the last term should read $\delta_{\alpha\beta} - \hat{q}_{\alpha} \hat{q}_{\beta}$, not just $1 - \hat{q}_{\alpha} \hat{q}_{\beta}$ (i.e., 1 should be substituted with $\delta_{\alpha\beta}$).

10. Below Eq. (3), you present an expression of $\omega_L^2(q)$ which seems to be taken at $q=0$, i.e., it looks like $\omega_L^2(0)$.

11. In your analysis, you took into account the shift of the LO\TO frequencies due to finite electron density according to Eq. (6). Is this the only effect of electrons on phonons and the interaction they mediate that you consider? Wouldn't it just imply a trivial position shift of the QCP?

12. How exactly do you derive Eq. (13)? What's ω_T appearing there, is it the bare one or renormalized according to Eq. (6)? If it's the bare one, where is the dependence on the electron's density? Also, right after Eq. (13) you mention "...since the integral is logarithmically divergent,..." What integral are you talking about there?

13. What's the difference between V_{En} in Eq. (13) and Eq. (11)? How (and why) are they different from V_{En}^{Pair} in Eq. (12)?

14. What's the definition of $\bar{\alpha}$ in Eq. (14)?

15. How exactly do you derive Eq. (14)? What density range (from Fig. 2) do you consider when deriving it? In other words, what real space-time form of the effective interaction do you assume to derive it (which one of the three from Fig. 2)?

16. Below Eq. (15), you say that $\alpha \ll 1$ because of $\Omega_0 \gg \Omega_T$. Can you explain how exactly the former follows from the latter?

17. What exactly are the quartic interactions between the phonons you're talking about? Why do you discuss this higher-order effect only? What about loop corrections to the single-particle electron and phonon Green's functions?

18. What ω_T do you use to plot Fig. 3? Again, are these the bare TO frequencies or the ones renormalized at finite electron density according to Eq. (6)? If these are the bare frequencies (masses), what happens to the effective interaction and to superconducting T_c right at the "real" QCP at finite density, i.e., when $\omega_T(n_e)=0$? What's the form of the effective interaction in this case?

19. The bare electron's action presented in the supplement seems to disagree with the one from the main text. There's an extra minus sign (I believe the one in the main text is the correct one).

Response to the Referees

We thank both the referees for their careful reading of our manuscript, and for their many comments on the science therein. We are particularly appreciative of your detailed suggestions that have helped to clarify our presentation. Referee 1 acknowledges “**the novelty of this study,**” noting that we ‘**justify the two-phonon exchange as the pairing mechanism for superconductivity in dilute metals.**” Referee 2 writes that “**The study is timely, while the topic and the results are very interesting.**”

Because of the length of our detailed responses to the Reviewers, here we provide a summary of the key points of our work:

- In dilute quantum critical polar metals, the Coulomb repulsion is heavily screened and the soft polar mode decouples from the charge in the case of negligible spin-orbit effects (which we will assume).
- Experimentally the superconducting transition temperature is enhanced with proximity to the polar quantum critical point. Since the electrons do not interact directly with the polar modes, we propose that it is the energy density of the zero-point polar fluctuations that drives the electron-electron attraction.
- We find that this electron-electron attraction, mediated by energy fluctuations, overcomes the Coulomb repulsion at low charge densities close to the polar quantum critical point resulting in a superconducting instability
- Application of our ideas to doped SrTiO₃ (STO) shows good agreement with experiment in the superconducting regime with the appropriate density. Since this interaction is known to describe anomalous electron conduction in the normal state, our theory provides a unified approach to transport in two different temperature regimes of doped STO; furthermore it explains the observed phonon frequency shift with doping.
- We predict that this mechanism will be considerably enhanced in two-dimensional quantum critical polar systems that can be realized in epitaxial films.

Please find below our itemized replies to the Reviewers’ specific points. Please note that, unless otherwise stated, all equation numbers refer to those in this Response but there are only Figures in the main text, where all changes are marked in red.

Referee 1

The novelty in this study is that the authors find that the charge fluctuation in

$$H_{En} = g \int d^3x \rho_e(\mathbf{r})(\vec{P}(\mathbf{r}))^2 \quad (1)$$

can lead to this attractive interaction and thus justify the two-phonon exchange as the pairing mechanism for superconductivity in dilute metals.

Comment: We would like to clarify that the pairing in our approach is due to quantum critical fluctuations of the energy density of the local polarization $[(\vec{P}(\mathbf{r}))^2$ in (1)]. While perturbatively the resulting attraction can be attributed to two-phonon exchange, phonon nonlinearities will lead to the energy fluctuation exchange being different from two-phonon one. For $d = 3$ the corrections are logarithmic but, as we discuss, in $d = 2$ the quartic interactions between the critical phonons are relevant and must be included.

However, I have a few comments/questions for the authors to address:

1. The theory is supposed to describe “dilute quantum critical polar metals”. What parameter is tuned in order to achieve the polar quantum critical point (QCP)? In experiments of $\text{Ca}_x\text{Sr}_{1-x}\text{TiO}_{3-\delta}$ (NP 13, 643 (2017)) and doped BaTiO_3 (PRL 104, 147602 (2010)), doped electrons suppress the polar displacements and lead to a polar QCP at a finite carrier density n_e . However, in the current theory, the transverse polar phonon mode has dependence on n_e as:

$$\omega_T(n_e) = \sqrt{\omega_{T0}^2 + gn_e\epsilon_o\Omega_0^2} \quad (2)$$

where $\omega_{T0} > 0$ is the transverse polar phonon frequency when mobile electrons vanish. From Eq. (2), does that mean $g < 0$ so as to achieve a polar QCP at a finite n_e ? However, if $g < 0$, then where is the polar QCP? I suppose that $\omega_{T0} > 0$ is a constant throughout the calculation. Clarification is needed here.

Reply: We certainly agree that in $\text{Ca}_x\text{Sr}_{1-x}\text{TiO}_3$ and in doped BaTiO_3 , the polar mode is observed to harden with electron density; similar effects have been observed in SrTiO_3 . It is for this reason that we take the coupling constant g to be positive in (2). Here ω_{T0}^2 refers to the undoped case; it is a bare quantity in the Lagrangian that measures the curvature of the free energy with respect to the transverse polarization. Different curves in Fig. 3 correspond to different values of $\omega_{T0}^2 \geq 0$. However we assume that at all the densities considered the

transverse optical phonon energy $\omega_T(n_e)$ is small with respect to Ω_0 and to Ω_T , leading to respectively a large lattice dielectric constant and to a large logarithmic enhancement of the attraction in Eq. (11) of the main text; this enhanced attraction originates from quantum fluctuations of the phonon energy density and therefore our theory does indeed include quantum critical phenomena.

If $\omega_{T0}^2 < 0$ in the bare Lagrangian, the system is polar and subsequent doping leads to $\omega_T^2(n_e) > \omega_{T0}^2$ if $g > 0$. This is indeed the situation for the two materials that you mention, $\text{Ca}_x\text{Sr}_{1-x}\text{TiO}_{3-\delta}$ and doped BaTiO_3 ; the insulating systems are initially polar ($\omega_{T0}^2 < 0$), and there is a true quantum phase transition at finite doping (see Nat. Phys 13, 643 (2017)). As mentioned briefly in the text, it would be interesting to generalize our theory to the ordered (polar) phase. On general grounds, order parameter and energy fluctuations are also expected to be prominent in the ordered phase close to the transition.

In the course of our revision, we found that a factor of two had to be included in Eq. (6) of the main text; this equation and its results displayed in Figure 3 have now been updated. We found no appreciable difference for the lower density range of our focus. However the results are somewhat different at higher densities, where the Matsubara frequency-dependence of both the screened Coulomb repulsion and the energy fluctuation-mediated attraction have to be included. A discussion of this point has been added on page six of the revised manuscript. **Action:** We have addressed and clarified these points in the revised text - see the discussion after Eq. 6 in the main text and in the middle of the left-hand column on page 6.

2. In Fig. 2, the authors show that the effective pairing potential V_{En}^{Pair} may have three different forms as the carrier density n_e changes. For low carrier density (middle region),

$$V_{En}^{Pair}(\mathbf{r}, \tau) \sim \frac{\delta(\tau)}{|\mathbf{r}|^3}. \quad (3)$$

For high carrier density (rightmost region),

$$V_{En}^{Pair}(\mathbf{r}, \tau) \sim \frac{\delta(\mathbf{r})}{\tau}. \quad (4)$$

For intermediate density (leftmost region),

$$V_{En}^{Pair}(\mathbf{r}, \tau) \sim \delta(\mathbf{r})\delta(\tau) \log(\xi a_0). \quad (5)$$

There are a few confusing points that need to be clarified:

2 (a): *The x-axis of Fig. 2 is monotonic with the carrier density. Then why does the leftmost region correspond to the “intermediate density”?*

Reply: We acknowledge the referee’s point and thank him/her for pointing out our error. In the main text, we mislabelled the low-density regime as the intermediate doping one. We apologize for the resulting confusion and this point has been corrected.

2 (b): *The paper wrote “For a low density polar metal that is critical at zero doping $\omega_T(n_e = 0) = 0$ ”, that means $\omega_{T0} = 0$. Is that a necessary condition to derive Eq. (2)? What happens if $\omega_{T0} > 0$?*

Reply: The necessary condition for the derivation of (2) is $c_s k_F \gg \omega_{T0}$.

Action: We have added a clarification of this point after Eq. (12) in the main text.

2(c): *For a finite $\omega_{T0} > 0$, since $2c_s k_F \sim n_e^{1/3}$ and $E_F \sim n_e^{2/3}$, when n_e is sufficiently small, $\omega_T \rightarrow \omega_{T0}$, which must exceed $2c_s k_F$ and E_F . Then does Eq. (3) correspond to ultralow densities, rather than “intermediate density”?*

Reply: We thank the referee for pointing out this misstatement. Yes, Eq. (3) refers to ultralow densities. We apologize for the confusion, and have corrected this point in the revised text.

Action: We have addressed the points raised in this set of questions in the revised text with clarifying phrases in the paragraph after Equation (11) in the main text; in particular please see the new caption of Fig. 2.

3. *When the authors apply their theory to doped SrTiO₃, can they estimate what the two critical carrier concentrations are in Fig. 2, given the experimental information (e.g. c_s , ϵ_1)? If it is doped BaTiO₃, whether these two critical carrier concentrations will be very different?*

Reply: The densities in Figure 2 describe crossovers between different interaction regimes of the quantum critical polar metal. For STO, we estimate that the lower and higher density crossovers occur around $n_1 \sim 1.3 \times 10^{17}$ and $n_2 \sim 2.6 \times 10^{19}$ respectively; note that we have modeled the “intermediate density” region when fitting to the three-dimensional STO data in Figure 3a. These distinct densities depend on several parameters that are material-specific; in particular, $n_1 \approx \frac{1}{3\pi^2} \left(\frac{\omega_{T0}}{2c_s} \right)^3$, $n_2 = 3\pi^2 \left(\frac{4m^*c_s}{\hbar} \right)^3$.

In the absence of doping, at ambient pressure BaTiO₃ has a high temperature ferroelec-

tric transition so that $\omega_{T0}^2 < 0$. Suppression of the polar transition temperature with doping has been reported, with T_C going to zero at $n^* \approx 1.9 \cdot 10^{21} \text{ cm}^{-3}$ [PRL 104, 147602 (2010)]. Taking the effective mass in BaTiO₃ to be around 10 electron masses [PRB 78, 045107 (2008)] and c_s around 1.5 times larger than in SrTiO₃ [$c_s^2 = 4750 \text{ (meV\AA)}^2$ from PRB 4, 155 (1971)], we find that $n_2 \approx 1.5 \cdot 10^{21} \text{ cm}^{-3}$. Therefore, at the QCP this system is already in the right-hand region of Fig. 2 where retardation effects are important. The polar transition can be additionally suppressed by pressure [Phys. Rev. Lett. 78, 2397 (1997)], shifting the QCP to lower densities and transforming BaTiO₃ into a dilute quantum critical polar metal. **Action:** We have included the numbers for the crossover densities in the revised text (see beginning of "Superconductivity in Doped SrTiO₃" section) and we have remarked in the penultimate paragraph that doped BaTiO₃ under pressure is another interesting candidate material.

4. (a) While Eq. (1) is allowed by symmetry, what is the microscopic origin of this term? In particular, what does g depend on? Is it strongly material-dependent? For example, in doped SrTiO₃, the authors use $g/a_0^3 = 0.62$. Then how about doped BaTiO₃ or other doped semiconductors?

Reply: The addition of negative charge to the conduction electron site attracts the displaced ions, increasing the "spring constant" between positive and negative charges. Microscopically, in STO and BTO the electrons are doping at the site of the undisplaced Ti ions, increasing the restoring forces on the displaced ones. Naturally g depends on the detailed Wannier function of the electrons and is thus material-dependent; however since its natural units are that of the unit cell volume, it is not expected to vary very much.

While there is no detailed neutron data available for doped BaTiO₃, one can estimate the coupling constant g from the linear rate of the suppression of the polar transition temperature [$\frac{dT_C(n)}{dn} = -19.4 \cdot 10^{-20} \text{ K} \cdot \text{cm}^3$ from the data in PRL 104, 147602 (2010)]. The transition temperature is determined by the softening of the TO phonon frequency; this can be described by a linear Curie-Weiss like expression $\omega_T^2(T, n) = \alpha(T - T_C(0)) + 2gn\varepsilon_0\Omega_0^2$, where $\alpha \approx 0.24 \text{ meV}^2/\text{K}$ and $\Omega_0 \approx 200 \text{ meV}$ using LST relation $\varepsilon(T) \approx \frac{\Omega_0^2}{\omega_T^2(T, n=0)}$ [Phys. Rev. B 26, 5904 (1982)]. The actual transition temperature is determined by $\omega_T^2(T, n) = 0$ and one can thus estimate $g = \frac{dT_C(n)}{dn} \frac{\alpha}{2\varepsilon_0\Omega_0^2}$. Using the BaTiO₃ lattice constant $a_0 \approx 0.4 \text{ nm}$, one gets $g/a_0^3 \approx 0.12$.

Action: We have added brief remarks discussing the microscopic origin of the interaction in the main text after Eq. (1).

4. (b) A more general question is: what is the condition under which Eq. (5) becomes

the dominant interaction in addition to Coulomb repulsion? Is it that the material must be close to the polar QCP? Or is it that the carrier density must be low? Or both?

Reply: In short we need both conditions: proximity to the QCP so that the electron-electron interactions are heavily screened, and low density so that the attraction due to energy fluctuations is large enough to overcome the Coulomb repulsion. In particular, Eq. (13) of the main text implies suppression of the attractive interaction with increase in the density-dependent quantities k_f and ω_T ; the latter set the scale of the momentum transfer and the distance from the QCP respectively.

Action: We have emphasized the need for both conditions more clearly in the revised text (p.4 after Eq. (16))

5. *The presentation needs a substantial improvement. This is a pure theoretical modelling. However, the notations are sometimes confusing. An incomplete list include:*

5. (a) In the main text, Eq. (11), (13) and (18), what does $i\omega$ mean? Is ω a real frequency or a Matsubara frequency? Eq. (17) uses $i\omega_n$. Do they mean the same thing?

Reply: ω should have been written as ω_n - a Matsubara frequency. We have corrected this and removed other notational ambiguities in the revised version.

5. (b) In the main text, Eq. (16) n_{max} has the dimension $1/[\text{length}]^3$. Then what does the text $n_{max} \ll 1$ mean?

Reply: We thank the referee for pointing out this oversight. Here it should have been $n_{max}a_0^3$ and this has been corrected.

5. (c): In the main text, we have n_e and n . Do they mean the same thing (i.e. carrier density)? If yes, can we just use one symbol to avoid confusion?

Reply: Yes, they are indeed both the electron density; we have replaced n with n_e throughout the text.

5. (d) In the main text, we have $\omega_T(\mathbf{q})$, $\omega_t(n_e)$, and a constant ω_T . In the SI, we have $\omega(n_e, \mathbf{q})$. In the derivation, we have $\omega_T \gg \omega_n$. What does that ω_T mean? It means the

constant term (i.e. $n_e = 0$ and $\mathbf{q} = 0$), or a more general $\omega_T(n_e, \mathbf{q})$?

Reply: Again we apologize for the ambiguous notation and we have tidied it up in the revised text. In particular, for each quantity we have indicated its dependent variables. Thus the optical phonon energy $\omega_T(n_e, \vec{q})$ depends on both the electron density n_e and momentum \vec{q} . $\omega_{T0} \equiv \omega_T(n_e = 0)$ is the transverse phonon energy at $\vec{q} = 0$ and $n_e = 0$. At sufficiently low densities ($n_e \ll \frac{\omega_{T0}^2}{2g\varepsilon_0\Omega_0^2}$; leftmost side of Fig. 2) $\omega_T(n_e) \approx \omega_{T0}$. Finally, $\Omega_T = \max_{\vec{q}} \omega_T(n_e, \vec{q})$ provides the large-energy cutoff for the phonon energies.

Action: We have made the notation more transparent and consistent in the revised text and the Supplementary Material.

5. (f) In the SI, Eq. (1) reads $S = S_e + S_{ph} + S_{e-ph}$. However, S_{e-ph} is not defined at all. Why not read $S = S_e + S_{ph} + S_{En} + S_{Coul}$ explicitly?

Reply: We thank the Referee for pointing out this inconsistency - in fact, this is a typo, which we have corrected in the new version. Indeed S_{En} is meant instead of S_{e-ph} .

5. (g) Sometimes a vector is denoted as an arrow on top of a letter and in other occasions is denoted as bold font. Is there a particular reason for this?

Reply: We have unified the vector notation in the revised text.

Action: We have streamlined our notation throughout the revised main text and in the Supplementary Information to address the points from this set of questions.

6. Finally I have a comment: the so-called “dilute quantum critical polar metal” is very rare. Probably doped SrTiO_3 is the only known example. SrTiO_3 is unique in that the material by itself is in the vicinity of ferroelectric phase transition. In other polar metals (such as LiOsO_3 or doped ferroelectric materials (such as doped BaTiO_3), while it is possible to drive them close to a polar QCP via strain or doping, usually the carrier density around the polar QCP is reasonably high ($10^{21}/\text{cm}^3$) and therefore the Fermi surface is sufficiently large so that the conventional electron-phonon mechanism is working. Therefore I am wondering whether the current pairing mechanism can be applied to any material system other than doped SrTiO_3 ?

Reply: We thank the Referee for the opportunity to discuss this important point. There are good reasons to believe that our theory applies to several polar metals and thus not just

to STO; these include other doped perovskites such as BaTiO₃ and KTaO₃. In the case of BaTiO₃, one can realize a dilute quantum critical polar metal by suppressing polar order with pressure [PRL 78, 2397 (1997)] at low carrier concentrations. Alternatively one can suppress ferroelectricity in insulating BTO by pressure [PRL 78, 2397 (1997)] and realize the polar metallic QCP at arbitrary low densities. By contrast, the paraelectric KTaO₃ is further from the QCP than is SrTiO₃ with $\omega_{T0} \approx 2.5$ meV (Phys. Rev. B 51, 8046 (1995)). Thin films of *KTaO₃*, however, do display ferroelectricity (Appl. Phys. Lett. 99, 052908 (2011)) due to lattice mismatch strain. Using tunable strain (Rev. of Sc. Inst. 85, 065003 (2014)) one can therefore bring KTaO₃ to the QCP at arbitrary low carrier concentrations.

Second, our ideas may be applicable to interfaces of these materials, in particular to LaAlO₃/SrTiO₃. Indeed, both the dome-shape of T_c as well as the comparable maximal T_c value (APL Mater. 4, 060701 (2016)) suggest that pairing mechanism is the same as in bulk SrTiO₃ (PRB 96, 094518 (2017)). Detailed application of our mechanism to LaAlO₃/SrTiO₃ interfaces requires serious consideration of band reconstruction effects beyond the scope of this work.

Another interesting candidate is LiOsO₃. This material is believed to have nodal points and lines close to the Fermi level [PRMats 2, 051201(R) (2018)], which on doping can give rise to small Fermi pockets. Furthermore, a strain-tuned polar QCP has been also predicted [J. Phys.: Condens. Matter 32 125501 (2020)], making this material a candidate for our mechanism.

Finally, the attraction mechanism based on interaction Eq. (1) of the main text can be easily generalized to energy fluctuations of any other order parameter. Therefore, as we have mentioned in the revised text, it can provide an additional contribution to electron-electron attraction close to other types of structural instabilities (such as the A15 compounds); it may also play a role in other correlated electron systems where the importance of energy fluctuations has been identified.

Action: We have elaborated on the applications of our theory to materials beyond doped SrTiO₃ in the Outlook section in the revised manuscript.

Referee 2

The study is timely, while the topic and the results are very interesting. Although I think the paper deserves publication in some form, I believe the presentation can be improved significantly. I'll try to list the most important aspects that bothered me:

1. *The first important step in your study is the calculation of the effective electron-electron attraction mediated by the two-phonon processes at different dopings. Depending on the*

doping and the proximity to the critical point, the behavior is very different. Could you summarize all the results for different regimes in a concise reader-friendly manner? I believe Fig. 2 is supposed to help with that but I find it hard to understand its meaning properly. For example, what are the densities that correspond to the vertical grey dashed lines on that figures? Is there a regime corresponding to Eq. (10)? When you compare ω_T to $c * k_F$, do you have in mind the bare TO phonon mass or the one renormalized according to Eq. (6)? What's the meaning of different curves in Fig. 2?

Reply: The curves in Fig. 2 represent three energy scales, $\omega_T(n_e)$, $2c_s k_F$ and E_F in this problem, each of which are density-dependent. The effective form of the electron-electron interaction is determined by the energy scale that is dominant, and this is what we aimed to convey in Figure 2. In momentum space, this is seen directly from Eq. (11) [or Eq. (13) for the detailed form], where the momentum and energy are set to $2k_F$ and to E_F respectively, scales relevant for Cooper pairing; here all equations refer to the main text.

Let us now consider this effective interaction in real space. There are three density regions, related by crossovers, defined by the dominant energy scales involved. In Region I of low-density, $\omega_T(n_e) \sim \omega_{T0}$, is the dominant scale; there is no q - or ω -dependence, so the Fourier transform of Eq. (11) is a delta function in both space and imaginary time. $2c_s k_F \sim n_e^{\frac{1}{3}}$ is dominant in Region II; here ω_n -dependence in (11) can be neglected, but not \vec{q} leading to an interaction that is only local in time, but not space. Finally in Region III, $E_F \sim n_e^{\frac{2}{3}}$ so the absence of q -dependence indicates the interaction is local in space but not in time. Our study focuses on Region II, that of intermediate density, where the pairing interaction is local in time but not in space. We note that when applied to SrTiO₃ it is the region where the usual electron-phonon interaction can be firmly ruled out due to the low value of the Fermi energy compared to longitudinal phonon frequency.

Action: We have revised the text, Figure 2 and its caption to clarify the density-dependence of the effective pairing interaction.

2. You have introduced too many similar notations having different meaning. It makes it really hard to navigate across the paper. For example, there's $\omega_T, \omega_T(n_e), \omega_T(q), \omega_q, \Omega_T, \Omega_0$ all meaning different things.

Reply: We apologize for the notational redundancy and for any confusion it may have caused. In the new version we have indicated the dependent variables of each quantity. The optical phonon energy depends on both the electron density and the momentum $\omega_T(n_e, \vec{q})$. At low momenta, one has $\omega_T^2(n_e, \vec{q}) \approx \omega_T^2(n_e) + c_s^2 \vec{q}^2$. We ignore the effects of renormalization of c_s with changing electron density, as these effects are absent to lowest order in the small

coupling g (see Eq. (6) of the main text). $\omega_{T0} \equiv \omega_T(n_e = 0)$ is the transverse phonon energy at $\vec{q} = 0$ and $n_e = 0$. At sufficiently low densities ($n_e \ll \frac{\omega_{T0}^2}{2g\epsilon_0\Omega_0^2}$; left side of Fig. 2) $\omega_T(n_e) \approx \omega_{T0}$. Finally, $\Omega_T = \max_{\vec{q}} \omega_T(n_e, \vec{q})$ provides the large-energy cutoff for the phonon energies.

Action: We have streamlined and clarified the notation of the different frequencies we have introduced.

3. *For some reason, you spend a lot of time discussing the analogy with the “dark matter”. Of course, we all love analogies between different concepts in physics, but this one seems a bit inappropriate. I’m not an expert in “dark matter”, but I don’t think people already know convincingly what that is and how exactly it interacts with more conventional matter. To me, the problem of the two-phonon mediated superconductivity is interesting enough on its own, just from the condensed matter perspective. Moreover, sentences like “We find that polar quantum criticality results in long-range “gravitational” interactions that mediate attraction between the electrons in an electromagnetically neutral background.” almost made me believe that the nature of this interaction is gravitational rather than electromagnetic, which is extremely confusing.*

Reply: The reason we chose this analogy is that in fact, dark matter is known to interact with baryons gravitationally (this is how it has been detected), i.e. via the stress-energy tensor. Furthermore, this is to highlight the universal character of interactions mediated by energy fluctuations, rather than fluctuations of, e.g. spin or charge. Of course, the ultimate origin of this phenomenon is electrostatic, but the long-wavelength, emergent physics is driven by fluctuations in the energy against a charge neutral background. We have deliberately made the distinction between energy fluctuation exchange and two-phonon exchange to highlight the potential differences due to nonlinearities and non-perturbative effects. In the $d = 3 + 1$ case, the corrections to the two-phonon exchange due to phonon nonlinearities are only logarithmically divergent, but in $d = 2 + 1$ as we have mentioned in the paper, the energy fluctuations pick up an anomalous dimension, and they are fully distinct from the perturbative two-phonon processes. In short, it is more accurate to characterize the mechanism as being that of energy fluctuations, rather than perturbative two-phonon processes.

Action: We have expanded this point in the revised text, see p.4 bottom of the right-hand column.

4. *Instead of discussing the relation to the “dark matter”, I’d suggest to discuss the similarities and differences with some other works on the same topic. Specifically, it’d be very inter-*

esting to see some comments regarding this recent preprint: <https://arxiv.org/abs/2106.09530>. The authors of that work also obtained some log enhancement of the effective interaction in the Cooper channel, see their Eqs. (7)-(8), which, however, seems different from your log in Eq. (14), since it doesn't depend on k_F (on electron density). I understand that this paper appeared after yours, but nevertheless I'd be happy to see your comments.

Reply: The work mentioned by the Referee has two important distinctions from ours. First, it concerns the regime of extremely low densities, corresponding to the region I in Fig. 2 of our paper. The dependence on electron density is taken there only perturbatively in $c_s q / \omega_{T0}$, see eq. (8) of that work. Therefore, the resulting T_c appears a growing function of density, rather than a dome, as is demonstrated in our Figure 3. Secondly, the Coulomb interaction was not explicitly addressed there. With regard to the application to SrTiO₃, the regime considered in that work corresponds to densities below about 10^{18} cm^{-3} . In this regime, superconductivity is observed only for oxygen annealed (but not Nb doped) samples, and, furthermore, no Meissner effect has been observed [Collignon et al., Annu. Rev. Cond. Mat. Phys. 10, 25 (2019)], indicating possibly filamentary superconductivity. By contrast superconductivity at roughly 10^{19} cm^{-3} , our density regime of focus, has been observed for all types of doping and does display a Meissner effect, indicating that it is a bulk phenomenon.

Action: We have noted this recent paper in the conclusion of our revised text.

5. You state above Eq. (7) that the Fermi liquid state is not destroyed even at the QCP based on the observation that the two-phonon coupling is irrelevant. But is it a sufficient condition? Can it happen that some loop diagrams lead to some crucial (possibly non-analytic) corrections which eventually destroy the Fermi liquid at the QCP (like, e.g., Landau damping in a more conventional coupling to a critical mode)? I'm not encouraging you to perform such an analysis in this paper, but rather curious whether this statement is sufficiently substantiated.

Reply: In the cases where the Fermi liquid is destroyed – e. g. for nematic or spin density wave fluctuations coupled to a Fermi surface - the interactions are in fact relevant and become marginal after Landau damping is included (see S. Sachdev's "Quantum Phase Transitions"). We know of no case where the interactions are irrelevant and non-analyticities lead to destruction of the Fermi liquid. Basically, the irrelevance of the interaction implies that there are no infrared singularities in diagrams of any order, and therefore one may expect the scaling form of the propagators to be same as that for free particles.

Action: We have referred to the known cases of non-Fermi liquid behavior on p.3, left

column of the revised text.

6. *The second paragraph of the introduction mentions the alternative mechanism for coupling to the TO mode through the spin-orbit coupling. Then, the last sentence of this paragraph , "However, this appealing idea encounters a difficulty, for the critical modes of a polar QCP are transverse optic (TO) phonons that decouple from the electrons at low momenta", looks a bit strange and confusing since it is exactly the spin-orbit coupling that allows to couple electrons to the TO phonons directly even at zero momentum. Also, it would be appropriate to mention here some earlier papers that discussed that type of coupling and its consequences for superconductivity: <https://journals.aps.org/prb/abstract/10.1103/PhysRevB.74.012501>, <https://journals.aps.org/prl/abstract/10.1103/PhysRevLett.115.207002> and <https://journals.aps.org/prx/abstract/10.1103/PhysRevX.9.031046>.*

Reply: We thank the referee for this point. We acknowledge this possibility in the new version of the text, and have referenced these papers. The question of whether spin-orbit coupling might provide a way out for STO is an active topic of current debate. A key appeal of the energy fluctuation picture is that it provides a unified understanding of: (1) shift of the phonon frequency with doping (2) the anomalous T^2 resistivity at low densities and (3) superconductivity, unlike the other mechanisms.

Action: We have revised the main text accordingly and have included the suggested references in the introduction of the revised manuscript.

7. *The first paragraph of the intro says "A challenge to this mechanism is posed by superconductivity in low carrier metals near polar quantum critical points (QCPs)." I believe it's meant to be "... in low carrier density metals..."*

8. *Above Eq. (2), the term S_e should contain the sum over k , not just \vec{k} (i.e., sum over frequencies is missing).*

9. *In Eq. (3), the last term should read $\delta_{\alpha\beta} - \hat{q}_\alpha\hat{q}_\beta$, not just $1 - \hat{q}_\alpha\hat{q}_\beta$ (i.e., 1 should be substituted with $\delta_{\alpha\beta}$).*

Reply: We thank the Referee for spotting these errors and we have corrected them.

Action: We have addressed these points in the revised text.

10. *Below Eq. (3), you present an expression of $\omega_L^2(q)$ which seems to be taken at $q = 0$, i.e., it looks like $\omega_L^2(0)$.*

Action: We have revised the notation to clarify this situation.

11. *In your analysis, you took into account the shift of the LO/TO frequencies due to finite electron density according to Eq. (6). Is this the only effect of electrons on phonons and the interaction they mediate that you consider? Wouldn't it just imply a trivial position shift of the QCP?*

Reply: Indeed this effect is just a shift of the QCP. However, in a real material, the bare ω_{T0}^2 is fixed while the density is changed, so one does have to take this into account. In addition to that, for the resulting instantaneous interaction between electrons (such as in regime II of Fig. 2), one has to include vertex corrections to the interactions. The expression for T_c we use actually includes these as is explained in Refs. 33,34 of the main text.

Action: We have expanded this point in the revised text - see discussion after Eq. (6) and p. 4, middle of right column.

12. *How exactly do you derive Eq. (13)? What's ω_T appearing there, is it the bare one or renormalized according to Eq. (6)? If it's the bare one, where is the dependence on the electron's density? Also, right after Eq. (13) you mention "...since the integral is logarithmically divergent,..." What integral are you talking about there?*

Reply: Here $\omega_T(n_e)$ refers to the renormalized frequency given by Eq. (6) of the main text. We are referring to the Fourier transform of Equation (8) (in the main text) which can be found in the Supplementary Material.

Action: We clarify the notation and this point in the revised text.

13. *What's the difference between V_{En} in Eq. (13) and Eq. (11)? How (and why) are they different from V_{En}^{Pair} in Eq. (12)?*

Reply: Equation (11) (in the main text) is an "intuitive" result based on the interaction at the QCP (Eq. (10)) whereas in Equation (13), the cutoffs and finite phonon energy have been taken into account explicitly. V_{En}^{Pair} in Eq. (12) is the approximate form of the effective interaction in real space in the Cooper problem for the regime II in Fig. 2. In this regime, ω_n can be neglected in Eq. (11),(13) and therefore the interaction becomes instantaneous.

Action: We have expanded our description of Equations (11) and (13) in the revised text,

and in particular in the caption of Fig. 2.

14. *What's the definition of $\bar{\alpha}$ in Eq. (14)?*

Reply: Here the bar indicates an average over the Brillouin zone. In the spirit of the Debye approximations it amounts to replacing the optical mode velocity c_s with an appropriate Brillouin zone average \bar{c}_s .

Action: We have clarified this point in the text.

15. *How exactly do you derive Eq. (14)? What density range (from Fig. 2) do you consider when deriving it? In other words, what real space-time form of the effective interaction do you assume to derive it (which one of the three from Fig. 2)?*

Reply: Equation (14) (main text) was obtained by averaging the Coulomb interaction (4) (main text) and the pairing interaction (13) (main text) over the Fermi surface, writing $k - k' = 2k_F \sin \theta$ and then averaging over $\sin \theta d\theta$. The density is assumed to be in regime I or II of Fig. 2, i.e. $E_F \ll 2c_s k_F, \omega_T(n_e)$. For more details please see equation (8) in the Supplementary Material.

Action: The derivation of (14) in the revised text has been clarified with a reference to the relevant equation in the Supplementary Material.

16. *Below Eq. (15), you say that $\alpha \ll 1$ because of $\Omega_0 \gg \Omega_T$. Can you explain how exactly the former follows from the latter?*

Reply: Here α should be replaced by γ so that this reads $\gamma \ll 1$. We apologize for this error and have corrected it.

Action: We have corrected this typographical mistake.

17. *What exactly are the quartic interactions between the phonons you're talking about? Why do you discuss this higher-order effect only? What about loop corrections to the single-particle electron and phonon Green's functions?*

Reply: The quartic interactions are given in Eq. (1) of the Supplementary Material and have the form $u \int d\tau d\vec{r} |\vec{P}(\vec{r}, \tau)|^4$. These interactions can be present in the system inde-

pendently from energy fluctuation coupling and therefore can be important. In particular, in $d = 2 + 1$ they do affect the phonon Green's function (see Ref. 35) introducing an anomalous dimension. We use this to argue that the Fermi liquid is preserved and g is irrelevant in the main text, and we elaborate in the Supplementary Material. In $d = 3 + 1$ only weak logarithmic corrections are present due to u , which we neglect. We do assume g to be small so that self-energy effects due to g can be neglected provided that the coupling g is irrelevant (which we demonstrate to be true in the Supplementary Material).

Action: The discussion of quartic interactions in the 2D case has been clarified to address this point more extensively.

18. *What ω_T do you use to plot Fig. 3? Again, are these the bare TO frequencies or the ones renormalized at finite electron density according to Eq. (6)? If these are the bare frequencies (masses), what happens to the effective interaction and to superconducting T_c right at the "real" QCP at finite density, i.e., when $\omega_T(n_e) = 0$? What's the form of the effective interaction in this case?*

Reply: We use $\omega_T(n_e)$ given by Eq. (6) [i.e. renormalized ones], with the value of g given in the caption and ω_{T0} indicated in the panels. Since $g > 0$ and $\omega_{T0}^2 \geq 0$, the system does not have a finite-density QCP. Generally, in case $\omega_T(n_e) = 0$, the interaction would be the one corresponding to regime II or III in Fig. 2, depending on the density. This can be deduced from Eq. (13) that is valid for all regimes.

Action: We have streamlined our frequency notation in the revised text and have also provided more discussion of Equation (6) (main text)

19. *The bare electron's action presented in the supplement seems to disagree with the one from the main text. There's an extra minus sign (I believe the one in the main text is the correct one).*

Reply: We thank the Referee for spotting this inconsistency and we have corrected it.

Action: We have rationalized our signs between the Supplementary notation and the Main text.

Again we thank the Reviewers for their careful consideration of our paper, and we hope that they will agree that we have addressed their concerns.

REVIEWER COMMENTS

Reviewer #1 (Remarks to the Author):

The authors have addressed all my comments, in particular, clarified that their theory applies to materials that are already close to the polar QCP without electron doping. And adding electron doping hardens the polar mode and takes the material away from the polar QCP. This is different from other theories which describe an actual phase transition at finite doping.

I have one final comment for the authors to address before recommending publication. The authors proposed that their theory can be applied to many other materials systems (such as doped BaTiO₃ and KTaO₃, and polar metal LiOsO₃). Experimentally superconductivity has been found in bulk KTaO₃ and recently at KTaO₃ (111) surface. My question is whether there is an experimental smoking gun (e.g. gap/T_c ratio) that can show that the observed superconductivity arises from the pairing mechanism proposed in the current theory, rather than from the conventional electron-phonon mechanism, or other related theories (such as Ref. 8, 17, 23, 24 etc.)?

Reviewer #2 (Remarks to the Author):

Generally, I find the authors' reply and the changes made to the manuscript solid. Hence, I recommend this paper for publication. I would only appreciate if the authors explain me once again in more detail the difference between "energy fluctuations exchange" and "two-phonon mechanism". I see the authors emphasize it several times in their reply that these are actually different, though I still don't quite understand why. I always thought (maybe mistakenly) that the term "two-phonon mechanism" simply implies the coupling between electron density and the square of the lattice polarization, exactly like the one in Eq. (1) of the manuscript.

Reply to the Referees

We thank both Reviewers for their careful readings of our responses to their detailed comments. We are also grateful for their positive assessments of our revised manuscript and for Reviewer 2's recommendation that it should be published. Here we address their requests for clarification of specific points in their second reviews, noting that all equation, reference and figure numbers refer to the main text unless otherwise stated. We have accordingly expanded our discussion of these issues in our current manuscript, where all recent revisions are marked in red.

Reply to Referee 1

Referee 1 asked us to clarify experimental consequences that distinguish our proposed two-phonon mechanism for quantum critical polar superconductivity from others previously proposed in the literature:

“The authors proposed that their theory can be applied to many other materials systems (such as doped BaTiO3 and KTaO3, and polar metal LiOsO3). Experimentally superconductivity has been found in bulk KTaO3 and recently at KTaO3 (111) surface. My question is whether there is an experimental smoking gun (e.g. gap/Tc ratio) that can show that the observed superconductivity arises from the pairing mechanism proposed in the current theory, rather than from the conventional electron-phonon mechanism, or other related theories (such as Ref. 8, 17, 23, 24 etc.)?”

Reply: There are a number of key features of the energy fluctuation mechanism that distinguish it from other theories:

- For dilute quantum critical polar metals, a T^2 dependence of resistivity related to two-phonon scattering is a key indicator of a dominant coupling to energy fluctuations (Refs. 28,29).
- The result that the doped and undoped phonon frequencies are related by a simple expression involving the coupling constant and the charge density

$$\omega_T^2(n_e) - \omega_{TO}^2 \propto gn_e$$

(see Eq. 6) naturally describes the suppression of the polar state with doping, a feature that is observed experimentally (see References 15 and 33), but has not been explained by other mechanisms.

- The scaling of T_c with phonon frequency, discussed in the text surrounding Eq. 17, is another defining signature

$$\frac{d \ln T_c}{d \ln \omega_T(n_e)} \propto -1 / \log^2 \frac{\Omega_T}{\omega_T(n_e)}$$

for $c_s k_F \ll \omega_T(n_e)$ and

$$\frac{d \log T_c}{d \log \omega_T(n_e)} \propto -\log \left(\frac{c_s k_F}{\omega_T(n_e)} \right) \frac{\omega_T^2(n_e)}{(c_s k_F)^2}$$

for $c_s k_F \gg \omega_T(n_e)$. An important point here is that T_c is very sensitive to the TO phonon energy at the low doping concentrations we consider (see also Fig. 3).

- Finally, in typical quantum critical polar pairing scenarios (e.g. Annals of Physics, Volume 417,168142 (2020)), the normal state close to the polar QCP is expected to be a non-Fermi liquid. This is not the case in the energy fluctuation scenario since the coupling in Eq. 1 is irrelevant in the RG sense.

We now briefly contrast our theory with other proposed descriptions of superconductivity near polar QCPs. The interband-driven mechanisms (Refs. 23 and 24) require the Fermi energy to be close to a band crossing (such as a Dirac point), whereas Eq. 1 is allowed by symmetry even for a single band. Conventional electron-phonon coupling (Refs. 8, 11 and 17) does not lead to a shift of the phonon frequency with electron concentration and can be ruled out for low dopings, specifically where E_F is much smaller than the relevant phonon energies (unless additional assumptions are made, e.g. Journal of Superconductivity and Novel Magnetism 30, 845 (2017)). Note that the gap/ T_c ratio remains equal to the BCS value with energy fluctuation pairing (see first paragraph in “Superconductivity in Doped SrTiO₃” section), so this feature does not distinguish between our proposed and the conventional electron-phonon scenarios. Mechanisms based on spin-orbit coupling predict highly anisotropic (Ref. 21) or unconventional (Refs. 22,25 and 26) pairing, distinct from the isotropic s-wave pairing due to energy fluctuations discussed in our work.

Action: We have expanded our discussion of key signatures of our proposed energy fluctuation mechanism, contrasting them with predictions of other theories of quantum critical polar superconductivity in the literature.

Reply to Referee 2

Referee 2 asked us to provide additional clarification regarding the distinction between the "energy fluctuations exchange" and the "two-phonon mechanism":

"I would only appreciate if the authors explain me once again in more detail the difference between "energy fluctuations exchange" and "two-phonon mechanism". I see the authors emphasize it several times in their reply that these are actually different, though I still don't quite understand why. I always thought (maybe mistakenly) that the term "two-phonon mechanism" simply implies the coupling between electron density and the square of the lattice polarization, exactly like the one in Eq. (1) of the manuscript."

Reply: The distinction between the energy fluctuation and the perturbative two-phonon exchange mechanisms appears in the emergent electron-electron interaction $V_{En}(x - x')$ and becomes important close to criticality, particularly in lower dimensions. Indeed, there the non-perturbative corrections that emerge are precisely the anomalous scaling dimensions of the energy density that appear in the specific heat at a Wilson-Fisher fixed point. The one-loop expression for $V_{En}(x - x')$ indeed corresponds to the exchange of two phonons $\langle \vec{P}^2(x) \vec{P}^2(x') \rangle = \text{Tr}[D(x - x')^2]$, where $D(x - x') \sim \frac{1}{|x - x'|^{d-1}}$ is the bare phonon Green's function and d is the spatial dimension. However, this expression ignores possible nonlinear effects of the phonon-phonon interaction u (see Eq. (1) of the Supplementary Material).

In the long wavelength (small momentum $q \rightarrow 0$) limit near a polar QCP, the scaling of the interaction u will lead to singular corrections; in this case individual phonons will not be well-defined as quasiparticles and the behavior of $\langle \vec{P}^2(x) \vec{P}^2(x') \rangle$ will be determined by the universality class of the transition. In the current work we assumed that the relevant momenta $q \sim k_F$ are large enough to be beyond the critical region, where the singular critical corrections can be ignored. In 3D they are logarithmic, while in 2D phonon quasiparticles are more seriously affected by their interactions u leading to an anomalous dimension $D(x - x') \sim \frac{1}{|x - x'|^{d-1+\eta}}$ in the long-wavelength limit (see Supplementary Section 2). Therefore, in the absence of well-defined phonon quasiparticles, it is more appropriate to attribute the interaction to energy fluctuation exchange, as energy of the lattice fluctuations remains a well-defined concept even when phonons are not.

In our work we do focus on the carrier density regime where the electron-electron interaction can indeed be approximated well by the perturbative two-phonon exchange, since a weak-coupling approach is appropriate. Still we believe that our theory provides a first step towards the study of electron pairing due to non-analytical energy fluctuations, which then

can be continued in future studies.

Action: We have expanded on the distinction between the energy fluctuation and the perturbative two-phonon mechanisms in the revised text. We have also added clarifying discussion about the breakdown of phonon quasiparticles near Eq. 18, indicating that in this case the energy fluctuation exchange approach is valid despite the absence of phonons.

Again we thank both Reviewers for their time and effort considering our paper, and we hope that they will agree that we have addressed their concerns.

REVIEWERS' COMMENTS

Reviewer #1 (Remarks to the Author):

The authors have addressed my final comment. Now I can recommend the publication of this work in Nature Communications.

Reviewer #2 (Remarks to the Author):

I recommend the paper for publication.